# Social support, distress, stress, anxiety, and depression as predictors of suicidal thoughts among selected university students in Bangladesh

**Sihab Howlader, Sumaiya Abedin, Md. Mosfequr Rahman** [ID] *

Department of Population Science and Human Resource Development, University of Rajshahi, Rajshahi, Bangladesh

* mosfeque@ru.ac.bd

**Data Availability Statement:** All relevant data are within the paper and its Supporting Information file.

## Abstract

This study examines the association of perceived social support with suicidal thoughts among a young adult sample of university students and estimates the degree to which perceived stress, distress, anxiety, and depression may explain the association. A cross-sectional survey was conducted from June to September 2022 among 642 students, selected using the probability proportional to size procedure, at a large university in Bangladesh. We used the Multidimensional Scale of Perceived Social Support (MSPSS) to measure support. The Impact of Events Scale-Revised (IES-R), the Perceived Stress Scale (PSS-10), the Generalized Anxiety Disorders (GAD-7) scale, and the Patient Health Questionnaire (PHQ-9) were used to assess distress, stress, anxiety, and depression. The association between social support and suicidal thoughts was examined using multivariable logistic regression models. Mediation analyses were carried out using the Karlson, Holm, and Breen (KHB) method. We found that perceived social support was associated with lower odds of suicidal thoughts after controlling for other variables (adjusted odds ratio = 0.94; 95% confidence interval = 0.92–0.96). Mediation analysis showed that, after adjusting for potential confounders, the mediating effects among the total effect of perceived social support on suicidal thoughts were 56.1%, 20.8%, 22.5%, 38.8%, and 50.9% for all mental disorders together and four specific adverse mental health outcomes, i.e., perceived distress, stress, anxiety, and depression, respectively. This study demonstrates that perceived social support reduces the likelihood of suicidal thoughts among university students, and mental disorders have a partial mediating effect on the association. Suicide prevention and mental health promotion efforts among university students should consider encouraging students to build and strengthen a strong social support network.

## Introduction

Suicide is one of the leading causes of death among young people worldwide [1], and the rising rates of suicide and self-harm in this population group have led to increased concern among

**Funding:** The authors received no specific funding for this work.

**Competing interests:** The authors have declared that no competing interest exist.

the public, policymakers, and clinicians [2,3]. Approximately 800,000 suicide deaths are reported globally every year [4], while more than 10,000 people in Bangladesh are estimated to die by suicide, with a rate of 7.3 per 100,000 people per year [5]. However, the prevalence of suicidal thoughts or attempts is much higher and more frequent than actual suicide deaths. Suicidal thoughts, which include recurring thoughts about death, ranging from a wish to not wake up in the morning to the idea that others would be better off if the individual died to even fleeting thoughts of suicide or specific plans for suicide [6], are regarded as the first step of suicidal behavior and one of the strongest predictors of suicide death [7]. Although suicidal thoughts can be screened for the prevention of suicidal deaths [8], it is often a neglected issue, specifically among young adults in developing countries. Understanding the social and psychological factors that have an influence on the development and amelioration of suicidal thoughts could inform suicide prevention strategies.

The term "perceived social support" refers to the subjective availability of care and assistance received from social relationships, which includes emotional support (such as expressions of empathy), instrumental support (such as help with household chores), and informational support (such as financial advice) that can come from a variety of sources, including friends or family [9]. It is one of the potential protective factors of suicidal thoughts that warrants consideration. During the adolescence and young adulthood periods, individuals experience major changes in social roles and responsibilities, such as living independently, entering the college or university, or job market, marrying, or engaging in romantic relationships [10]. The incidence of mental health problems is also common during these periods, especially in young adult populations [11,12]. Therefore, support may be especially important in these periods. Previous studies documented that social support is protective against suicide ideation [13], in particular for individuals experiencing risk factors such as psychological distress, depression, and anxiety [14,15]. Social support may also work indirectly in reducing suicide or suicidal behaviors by increasing other protective factors such as self-esteem [16]. Along with this empirical evidence, strong theoretical support for the aforementioned relationship is also documented. For example, feelings of belonging may increase in the presence of social support. The existence of social support is negatively correlated with suicide risk, according to Joiner's Interpersonal Theory of Suicide [17–19]. Social support may also mean that someone is there to help in coping with difficult life circumstances linked to psychopathology, which could reduce suicidal thoughts, behaviors, and actions.

In addition to social support as a protective factor, there are also risk factors for suicidal thoughts. Research across the globe has reported that mental disorders such as depression, anxiety, post-traumatic stress disorder (PTSD), and substance use significantly increase the risk of suicidal ideation and attempts [20–23]. However, research on the potential mediational effects of different mental disorders linking social support and suicidal thoughts is limited. A recent study documented that anxiety and depressive symptoms mediate the relationships between subjective support, objective support, support utilization, and suicide acceptability among Chinese university students [24]. Such a mediational role of mental disorders was also observed in other population groups. Another Chinese study among drug users reported that depression could potentially mediate the association between perceived social support and suicide attempts [16]. For example, a study among Spanish adults (18 years of age and older) documented that the relationship between social support and suicidal thoughts and behaviors is significantly mediated by emotional disorders such as anxiety and depression [25]. Although a number of studies among university students in Bangladesh documented that either social support or mental disorders have a significant effect on suicidal behaviors [26–28], none of these studies showed how symptoms of mental health could potentially mediate the relationship between social support and suicidal thoughts. Thus, we wonder whether and to what extent mental disorders (distress, stress, anxiety,

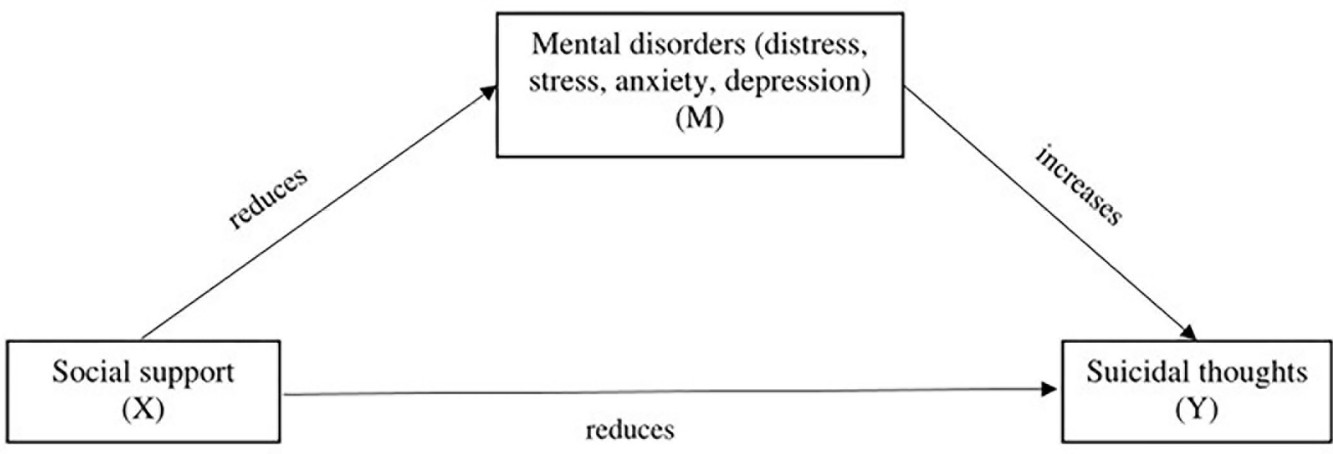

**Fig 1. Hypothesized framework of mediation analysis.**

and depression), individually and combined, play a mediational role in the association between perceived social support and suicidal thoughts among a young adult sample of university students. Therefore, this study examines the possible mediational role of distress, stress, anxiety, and depression in the association of perceived social support with suicidal thoughts among young adult university students in Bangladesh. In this study, we assume that: (i) a higher level of perceived social support is associated with fewer suicidal thoughts; (ii) mental disorders (distress, stress, anxiety, and depression) are positively associated with suicidal thoughts; and (iii) mental disorders (distress, stress, anxiety, and depression) mediate the association of perceived social support with suicidal thoughts (Fig 1). Understanding these complex relationships could be helpful in generating new ideas for suicide prevention strategies.

## Methods

### The survey

A cross-sectional study was conducted from June to September 2022, utilizing a paper-based survey among students of the University of Rajshahi, Bangladesh. The University of Rajshahi is the second-largest public university in Bangladesh, comprised of more than 33,000 students within 12 faculties and 59 departments. The primary sample of this study was adults aged 18 years and older who were currently enrolled at the undergraduate or graduate level in any department of the university. Based on the following assumptions: 19% of university students in Bangladesh reported having had suicidal ideation [29], a 95% confidence level and a 5% margin of error, the sample size was calculated using the single population proportion formula. The minimum required sample size for this study was 522 after taking into account a finite population correction, a design effect of 2, and a 10% non-response rate. However, a total of 700 students were invited to participate in this study during the study period; of these, 32 were absent, 18 were unable to attend due to schedule constraints, and 8 did not complete all the items of the questionnaire. This resulted in a final sample of 642 participants, yielding a response rate of 92%. Two-stage sampling procedures were used to collect the data. There are 12 faculties at the University of Rajshahi, with varying numbers of students. At the first stage, samples were allocated to each faculty, with the probability proportional to the size of the students. At the second stage, one department from each of the 12 faculties was chosen at random, and the assigned samples for each faculty were then distributed to each of the selected

departments. All students from the selected departments were eligible for this study. Finally, students were selected via systematic random sampling of their roll numbers. The objective, purpose, risks, and benefits of this study were fully explained to each participant. Prior to conducting the survey, written informed consent was obtained from the selected respondent. Interviews were conducted outside of class time. The survey was examined and got approval from the ethical review committee of the Institute of Biological Sciences, University of Rajshahi (Ref. 293(13)/320/IAMEBBC/IBSc).

## Variables

**Suicidal thoughts (outcome variable).** The outcome of interest was the students' suicidal thoughts, which was measured using a single-item question. Students were asked, "During the preceding month, did you seriously think about committing suicide?" with a response option of yes (= 1) or no (= 0).

**Social support (exposure variable).** Perceived social support was measured by the Multidimensional Scale of Perceived Social Support (MSPSS) [30]. The MSPSS is a 12-item, straightforward, and brief scale designed to subjectively assess the sufficiency of perceptions about social support. It is a 7-point Likert-type scale with response options ranging from 1 (very strongly disagree) to 7 (very strongly agree). The MSPSS estimates social support quality from three sources: family, friends, and significant others. Scores for all items were summed to compute a total score, with higher scores indicating greater perceived social support. The perceived social support scale showed high internal consistency in this study (Cronbach's alpha = 0.91).

## Potential mediators

**Distress.** The Impact of Events Scale-Revised (IES-R) [31,32], a self-reported 22-item scale, was used to measure the distress among the students resulting from a traumatic life event. Participants rated the extent to which each item applies to their experience during the past two weeks on a 0 (*not at all*) to 4 (*extremely*) scale. The score ranges from 0 to 88, with higher scores indicating higher distress levels. In our sample, the internal reliability of IES-R was adequate (Cronbach's alpha = 0.91).

**Stress.** The Perceived Stress Scale (PSS), a two-factor, 10-item self-reported instrument, was used to quantify stress. It is typically used to measure one's appraisal of life events as stressful [33]. Factors align with statements that are stated both negatively ($n = 6$) and positively ($n = 4$). Likert-type frequency response options were: *never* (= 0), *almost never* (= 1), *sometimes* (= 2), *fairly often* (= 3), and *very often* (= 4). The PSS total score ranges from 0 to 40 after reverse scoring for some items, with higher scores indicating increased stress. The PSS had adequate internal reliability in our sample (Cronbach's alpha = 0.85).

**Anxiety.** We assessed anxiety among the students using the 7-item Generalized Anxiety Disorders (GAD-7) scale. The items inquire about the degree to which the individual has been bothered during the last two weeks. Response options were: *not at all* (= 0), *several days* (= 1), *more than half the days* (= 2), and *nearly every day* (= 3). Items were coded in such a way that higher scores suggested a greater frequency of anxiety. The responses to all 7 items, with a total possible score ranging from 0 to 21, were added up to determine the student's level of anxiety (Cronbach's alpha = 0.93).

**Depression.** The Patient Health Questionnaire (PHQ-9) was used to assess depression [34], which assesses nine depressive symptoms occurring from *not at all* (= 0) to *nearly every day* (= 3) during the past two weeks. The score ranges from 0 to 27, with a higher score suggesting a greater likelihood of a major depressive disorder. In our sample, the internal consistency of the PHQ-9 was satisfactory (Cronbach's alpha = 0.79).

### Control variables

These included socio-demographic variables such as age, gender, academic status, place of residence, father and mother's educational attainment, father's occupation, mother's occupation, and number of family members.

### Statistical analysis

The statistical analysis for this study was conducted using Stata 16 (Stata Corp LP, College Station, Texas). The differences in the sample characteristics between students having suicidal thoughts or not were tested by Student's *t*-tests and Chi-squared tests for continuous and categorical variables, respectively. A series of multivariable logistic regression models were constructed to investigate the association between social support (exposure) and suicidal thoughts (outcome). In the first logistic model, suicidal thought was regressed only on social support, while control variables were added in the second model. Distress, stress, anxiety, and depression were included in models 3 to 6, respectively. Finally, in model seven, all variables were included (control variables and potential mediators: stress, distress, anxiety, and depression). We assessed models for multicollinearity.

We conducted a mediation analysis to examine the extent to which the association between social support and suicidal thoughts was mediated by mental health issues such as distress, stress, anxiety, and depression while controlling for sociodemographic variables. The KHB (Karlson, Holm, and Breen) approach was used in Stata to carry out the mediation analysis [35]. This analysis determined the contribution of each potential mediator, both individually and collectively, to the main association (social support predicting suicidal thoughts). The KHB method-based mediation analysis allowed us to deconstruct the total effect of social support on suicidal thoughts (i.e., unadjusted for the mediator) into the direct effect of social support on suicidal thoughts (i.e., adjusted for the mediator) and the indirect effect (i.e., the mediational effect) of social support on suicidal thoughts through mediators. This method was also used to calculate the mediated percentage, which is the portion of the main association that the mediators account for. This method is appropriate for estimating the effects (total, direct, and indirect) with nonlinear probability models such as those calculated using logistic regression. Each potential mediator (distress, stress, anxiety, and depression) was entered into the model individually to investigate their impact, and then all mediational variables were entered simultaneously to examine their combined effect. Results from the regression analyses are presented as odds ratios (ORs) with 95% confidence intervals (CIs). All reported probabilities (p-values) were two-sided, and <0.05 was considered statistically significant.

## Results

A total of 642 students (37.8% female) with a mean (SD) age of 22.0 (2.07) years (range: 18–27) were included in the analysis. The majority (60.2%) of the students were from rural areas, 28% had fathers with less than secondary education, and 38% had mothers with less than secondary education. The prevalence of suicidal thoughts in the past month among the students was 25.1%. The mean score (MSPSS) of social support reported by the students was 56.40 (SD:13.61) out of a possible range of 12 to 84. The average self-reported distress score was 36.02 out of a possible range of zero to 88, while the average perceived stress score was 16.62 out of a possible range of zero to 40. The mean score of anxiety computed by GAD-7 was 10.07 out of a possible range of zero to 21, while the average depression score measured by PHQ-9 was 12.74 out of a possible range of zero to 27 (Table 1).

Variables such as father's education, occupation, family size, perceived social support, and mental health issues (distress, stress, anxiety, and depression) were found to be associated with suicidal thoughts (Table 1). Students whose father had no or primary education were more

**Table 1. Sample characteristics (N = 642).**

| Characteristics | Mean (SD) / Number (%) | Suicidal thoughts n (%)/Mean (SD) | | p-value (t/$\chi^2$-test) |
| --- | --- | --- | --- | --- |
| | | Yes (n = 161) | No (n = 481) | |
| **Age (years)** | 22.0 (2.07) (Range: 18–27) | 22.20 (2.30) | 21.95 (2.00) | 0.188 |
| **Gender** | | | | 0.842 |
| Male | 399 (62.2) | 99 (24.8) | 300 (75.2) | |
| Female | 243 (37.8) | 62 (25.5) | 181 (74.5) | |
| **Academic status** | | | | 0.618 |
| Undergraduate student | 519 (80.8) | 128 (24.7) | 391 (75.3) | |
| Graduate student | 123 (19.2) | 33 (26.8) | 90 (73.2) | |
| **Area of residence** | | | | 0.423 |
| Rural | 388 (60.4) | 93 (24.0) | 295 (76.0) | |
| Urban | 254 (39.6) | 68 (26.8) | 186 (73.2) | |
| **Father's education** | | | | 0.004 |
| No education or Primary | 165 (25.7) | 53 (32.1) | 112 (67.9) | |
| Secondary | 141 (22.0) | 42 (29.8) | 99 (70.2) | |
| Higher | 336 (52.3) | 66 (19.6) | 270 (80.4) | |
| **Mother's education** | | | | 0.173 |
| No education or primary | 245 (38.2) | 65 (26.5) | 180 (73.4) | |
| Secondary | 217 (33.8) | 60 (27.6) | 157 (72.4) | |
| Higher | 180 (28.0) | 36 (20.0) | 144 (80.0) | |
| **Father's occupation** | | | | 0.002 |
| Farmer | 196 (30.5) | 40 (20.4) | 156 (79.6) | |
| Government service | 130 (20.3) | 25 (19.2) | 105 (80.8) | |
| Non-government service | 97 (15.1) | 21 (21.6) | 76 (78.4) | |
| Business or others | 219 (34.1) | 75 (34.3) | 144 (65.7) | |
| **Mother's occupation** | | | | 0.702 |
| Homemakers | 573 (89.3) | 145 (25.3) | 428 (74.7) | |
| Service or others | 69 (10.7) | 16 (23.2) | 53 (76.8) | |
| **Family size** | | | | 0.003 |
| 2–4 | 251 (39.1) | 53 (21.1) | 198 (78.9) | |
| 5–6 | 298 (46.4) | 72 (24.2) | 226 (75.8) | |
| >6 | 93 (14.5) | 36 (38.7) | 57 (61.3) | |
| **Social support** | 56.40 (13.61) (Range: 12–84) | 47.19 (12.73) | 59.49 (12.47) | <0.001 |
| **Distress** | 36.02 (16.21) Range (0–88) | 47.01 (12.99) | 32.35 (15.51) | <0.001 |
| **Stress** | 16.62 (7.67) Range (0–40) | 21.32 (6.18) | 15.04 (7.48) | <0.001 |
| **Anxiety** | 10.07 (6.33) Range (0–21) | 15.68 (5.24) | 8.18 (5.50) | <0.001 |
| **Depression** | 12.74 (7.51) Range (0–27) | 19.74 (5.93) | 10.40 (6.45) | <0.001 |
| **Total** | | 161 (25.1) | 481 (74.9) | |

likely to report having suicidal thoughts than students whose father had higher education (32.1% vs. 19.6%; p = 0.004). A higher prevalence of suicidal thoughts was reported among students with businessman fathers than fathers working for governments (34.3% vs. 19.2%; p = 0.002).

**Table 2. Association between social support, mental disorders and suicidal thoughts assessed by logistic regression.**

| Variables | Model 1 OR (95% CI) | Model 2 OR (95% CI) | Model 3 OR (95% CI) | Model 4 OR (95% CI) | Model 5 OR (95% CI) | Model 6 OR (95% CI) | Model 7 OR (95% CI) |
|---|---|---|---|---|---|---|---|
| **Social support** | 0.93 (0.92–0.95) | 0.92 (0.91–0.94) | 0.92 (0.90–0.94) | 0.93 (0.91–0.95) | 0.93 (0.91–0.95) | 0.94 (0.92–0.96) | 0.94 (0.92–0.97) |
| **Distress** | | | 1.07 (1.05–1.09) | | | | 1.03 (1.01–1.05) |
| **Stress** | | | | 1.15 (1.10–1.19) | | | 1.05 (1.00–1.10) |
| **Anxiety** | | | | | 1.30 (1.22–1.37) | | 1.11 (1.03–1.20) |
| **Depression** | | | | | | 1.30 (1.23–1.37) | 1.18 (1.10–1.27) |
| **Age (years)** | | 1.06 (0.93–1.21) | 0.99 (0.86–1.14) | 0.97 (0.84–1.12) | 0.91 (0.78–1.08) | 0.88 (0.73–1.04) | 0.82 (0.68–0.99) |
| **Gender** | | | | | | | |
| Male | | [reference] | [reference] | [reference] | [reference] | [reference] | [reference] |
| Female | | 1.02 (0.63–1.65) | 0.64 (0.38–1.10) | 0.84 (0.50–1.42) | 0.50 (0.27–0.97) | 0.50 (0.27–0.94) | 0.34 (0.18–0.68) |
| **Academic status** | | | | | | | |
| Undergraduate student | | [reference] | [reference] | [reference] | [reference] | [reference] | [reference] |
| Graduate student | | 1.16 (0.54–2.51) | 1.71 (0.76–3.85) | 1.61 (0.70–3.67) | 3.44 (1.27–9.31) | 2.41 (0.90–6.49) | 3.57 (1.26–10.10) |
| **Area of residence** | | | | | | | |
| Rural | | [reference] | [reference] | [reference] | [reference] | [reference] | [reference] |
| Urban | | 1.90 (1.04–3.45) | 2.49 (1.28–4.81) | 3.09 (1.59–6.01) | 4.08 (1.96–8.48) | 7.45 (3.28–16.90) | 10.82 (4.26–27.54) |
| **Father's education** | | | | | | | |
| No education or Primary | | [reference] | [reference] | [reference] | [reference] | [reference] | [reference] |
| Secondary | | 0.58 (0.29–1.16) | 0.46 (0.22–0.98) | 0.52 (0.25–1.10) | 0.53 (0.23–1.20) | 0.70 (0.31–1.61) | 0.60 (0.25–1.41) |
| Higher | | 0.21 (0.10–0.46) | 0.22 (0.09–0.51) | 0.21 (0.09–0.50) | 0.29 (0.12–0.71) | 0.39 (0.16–0.99) | 0.33 (0.12–0.88) |
| **Mother's education** | | | | | | | |
| No education or primary | | [reference] | [reference] | [reference] | [reference] | [reference] | [reference] |
| Secondary | | 1.38 (0.70–2.75) | 1.95 (0.96–3.96) | 1.93 (0.91–4.07) | 1.71 (0.79–3.71) | 0.71 (0.32–1.59) | 1.29 (0.56–3.00) |
| Higher | | 1.14 (0.51–2.54) | 1.04 (0.44–2.43) | 1.10 (0.47–2.58) | 0.83 (0.34–2.00) | 0.41 (0.16–1.04) | 0.50 (0.19–1.31) |
| **Father's occupation** | | | | | | | |
| Farmer | | [reference] | [reference] | [reference] | [reference] | [reference] | [reference] |
| Government service | | 1.30 (0.55–3.05) | 1.16 (0.45–3.00) | 1.16 (0.45–3.01) | 2.05 (0.71–5.88) | 1.31 (0.44–3.86) | 1.44 (0.47–4.40) |
| Non-government service | | 1.66 (0.75–3.70) | 2.02 (0.84–4.88) | 1.54 (0.66–3.58) | 1.66 (0.67–4.12) | 1.38 (0.53–3.64) | 1.38 (0.50–3.80) |
| Business or others | | 1.92 (1.09–3.39) | 2.52 (1.35–4.71) | 2.04 (1.10–3.80) | 1.75 (0.90–3.43) | 1.16 (0.57–2.32) | 1.39 (0.66–2.91) |
| **Mother's occupation** | | | | | | | |
| Homemakers | | [reference] | [reference] | [reference] | [reference] | [reference] | [reference] |
| Service or others | | 0.77 (0.35–1.68) | 0.67 (0.28–1.60) | 0.49 (0.21–1.14) | 0.49 (0.19–1.23) | 0.40 (0.16–1.03) | 0.29 (0.10–0.83) |
| **Family size** | | | | | | | |
| 2–4 | | [reference] | [reference] | [reference] | [reference] | [reference] | [reference] |
| 5–6 | | 0.68 (0.39–1.18) | 0.86 (0.47–1.58) | 0.86 (0.48–1.57) | 0.83 (0.44–1.59) | 0.63 (0.31–1.25) | 0.99 (0.48–2.05) |
| >6 | | 1.29 (0.61–2.74) | 1.59 (0.70–3.61) | 1.78 (0.79–4.01) | 1.74 (0.74–4.05) | 1.38 (0.56–3.37) | 1.74 (0.69–4.35) |

Table 2 represents the results from a logistic regression analysis with seven different models that explore the association between perceived social support and suicidal thoughts. Perceived social support significantly predicts the suicidal thoughts of the students, as it is statistically significant in all the models. A unit increase in the perceived social support scale decreases the odds of suicidal thoughts by 6% after controlling for all other variables and potential mediators (Model 7). All the mental health issues were also found to be significantly positively associated with suicidal thoughts when they were assessed individually as well as combined. For every one point increase in the distress, stress, anxiety, and depression score, it was associated with 3% (adjusted odds ratio [AOR] = 1.03; 95% confidence interval [95% CI] = 1.01–1.05), 5% (AOR = 1.05; 95% CI = 1.00–1.10), 11% (AOR = 1.11; 95% CI = 1.03–1.20), and 18%

**Table 3. Distress, stress, anxiety, and depression as mediator in the relationship between perceived social support and suicidal thoughts among university students in Bangladesh.**

| Mediator | Effect | β | SE(β) | z | p-value | OR (95% CI) | % Mediated* |
|---|---|---|---|---|---|---|---|
| All mediators | Total | -0.123 | 0.014 | -8.87 | <0.001 | 0.88 (0.86–0.91) | 56.1 (Distress 6.5 + Stress 5.7 + Anxiety 12.2+ Depression 31.7) |
| | Direct | -0.053 | 0.012 | -4.51 | <0.001 | 0.95 (0.93–0.97) | |
| | Indirect | -0.069 | 0.009 | -7.36 | <0.001 | 0.93 (0.92–0.95) | |
| Distress | Total | -0.096 | 0.011 | -8.93 | <0.001 | 0.91 (0.89–0.93) | 20.8 |
| | Direct | -0.076 | 0.010 | -7.65 | <0.001 | 0.93 (0.91–0.94) | |
| | Indirect | -0.020 | 0.004 | -5.12 | <0.001 | 0.98 (0.97–0.99) | |
| Stress | Total | -0.089 | 0.010 | -9.00 | <0.001 | 0.91 (0.90–0.93) | 22.5 |
| | Direct | -0.069 | 0.009 | -7.29 | <0.001 | 0.93 (0.92–0.95) | |
| | Indirect | -0.020 | 0.004 | -5.22 | <0.001 | 0.98 (0.97–0.99) | |
| Anxiety | Total | -0.103 | 0.011 | -8.94 | <0.001 | 0.90 (0.88–0.92) | 38.8 |
| | Direct | -0.062 | 0.011 | -5.87 | <0.001 | 0.94 (0.92–0.96) | |
| | Indirect | -0.040 | 0.006 | -6.89 | <0.001 | 0.96 (0.95–0.97) | |
| Depression | Total | -0.112 | 0.012 | -9.12 | <0.001 | 0.89 (0.87–0.92) | 50.9 |
| | Direct | -0.055 | 0.011 | -5.11 | <0.001 | 0.95 (0.93–0.97) | |
| | Indirect | -0.057 | 0.008 | -7.26 | <0.001 | 0.94 (0.93–0.96) | |

Total effect is the effect of social support on suicidal thoughts without considering mediator(s); Direct effect is the effect of social support on suicidal thought when controlling for either distress, or stress, or anxiety, or depression, or all mediators. Indirect effect is the effect of social support on suicidal thought through for either distress, or stress, or anxiety, or depression, or all mediators. *Mediation (%) is calculated by Indirect effect/Total effect x100. Adjusted for age, age, gender, academic status, place of residence, father and mother's educational attainment and occupation, family size.

(AOR = 1.18; 95% CI = 1.10–1.27) higher odds of suicidal thoughts among students, respectively.

Table 3 displays all the adjusted results of the KHB method-based mediation analysis. It is noteworthy to mention that the calculated odds ratios shown in Table 3 are on the same parameter scale and are therefore not comparable to the odds ratios reported in Table 2. The odds ratios shown in Table 2 are based on the outcomes of logistic regression for nested non-linear probability models, which have dissimilar scale parameters depending on the independent variables included in each model. The total effect of perceived social support on suicidal thought was an odds ratio of 0.88 (95% CI = 0.86–0.91). There remained a direct effect of perceived social support on suicidal thoughts independent of the potential mediators (OR = 0.95, 95% CI = 0.93–0.97). The odds ratio for the indirect effect of social support on suicidal thoughts through distress, stress, anxiety, and depression was 0.93 (95% CI = 0.92–0.95), indicating that there was a mediated effect with distress, stress, anxiety, and depression, which accounts for 56.1% (distress 6.5%, stress 5.7%, anxiety 12.2%, and depression 31.7%) of the total effect of social support on suicidal thought, accounting for all other covariates. As for four specific adverse mental health outcomes, after adjusting for potential confounders, the mediating effects on the association of perceived social support with suicidal thought were 20.8%, 22.5%, 38.8%, and 50.9% for distress, stress, anxiety, and depression, respectively.

## Discussion

This study estimates the prevalence of suicidal thoughts among university students and their relationship with perceived social support. This study further investigated the mediational effect of adverse mental health outcomes such as distress, stress, anxiety, and depression on the relationship between social support and suicidal thoughts. The prevalence of suicidal thoughts in the past 30 days is 25.1%, which is alarming because the reported prevalence is higher than

that documented in previous Bangladeshi studies [29,36,37]. The results of this study also revealed that perceived social support and suicidal thoughts were significantly associated. As the amount of perceived social support increases, university students are less likely to experience suicidal thoughts, supporting our first hypothesis. Results also indicated that increasing levels of distress, stress, anxiety, and depression were significantly associated with a higher likelihood of suicidal thoughts, supporting our second hypothesis. However, it is also found that suicidal thoughts among university students are not only directly affected by their level of perceived social support but that perceived social support also plays a protective role via distress, stress, anxiety, and depression, which supports the third hypothesis of this study. In other words, a higher perceived level of social support was linked to lower levels of distress, stress, anxiety, and depression, which in turn reduced the likelihood of suicidal thoughts.

These findings highlight the significance of perceived social support for the suicidal thoughts of university students. This is consistent with other studies outlining the relationships between perceived social support and suicidal thoughts and behaviors among other populations across the globe [16,38–40] including Bangladesh [41,42]. The findings of this study confirm some of the related fundamental theoretical works that focused on suicidal thoughts and behaviors related to social connectedness, such as thwarted belongingness and perceived burdensomeness, in the context of the interpersonal theory of suicide [18,19,43]. Our findings uphold the theory that, regardless of the level of stress a person may be experiencing at any given time, consistent, ongoing social support confers mental health benefits for individuals [44,45]. In accordance to this model, since university students are suffering from mental disorders at a higher rate after the COVID-19 pandemic [26], consistent feeling of support from family, friends, and significant others could help them deal with the circumstances that lead to better mental health and make them less likely to consider suicide. For university students, the university as a social context plays an important role in the students' social connectedness with peers, which can inform mental health practitioners regarding the risk of emotional distress. Along with academic stress, a lack of social support within the university could lead to an elevated risk of suicidal thoughts, regardless of their supportive environment at home.

We found that our proposed hypothetical models of mediation on the relationship between perceived social support and suicidal thoughts were confirmed with high significance levels. Since mental disorders, specifically emotional disorders (anxiety and depression), are significant risk factors for suicidal behaviors and suicide [21,46–48], the presence of these disorders would enhance suicidal thoughts among university students. Therefore, perceived social support affects students' suicidal thoughts by reducing the level of mental disorders. University students particularly encounter challenges due to the transition to adulthood, the demanding academic environment, and the experience of financial strain [49–51], which might lead them to endure higher degrees of psychological distress than the general population. For example, a study reported that being a student was significantly associated with an increased risk of developing depressive symptoms compared to individuals with other occupational statuses (such as employment or retirement) [52]. Similarly, the associations between suicidal thoughts and academic stress [53] or perceived social support [54] were found to be mediated by depression. Social support is therefore one of the essential elements in understanding the symptoms of mental health, which are linked to suicidal thoughts. This empirical model supports theoretical viewpoints that highlight interpersonal factors as prominent deterrents of suicidal thoughts and behaviors [55,56].

It is imperative to acknowledge the potential limitations inherent in this study. First, the utilization of cross-sectional data imposes limitations on our ability to examine the directionality of the hypotheses. For example, social support has the potential to exert an influence on suicidal thoughts. However, it is also plausible that suicidal thoughts or adverse mental outcomes may also have an influence on social support. It is suggested that future longitudinal research

be conducted in order to ascertain the temporal nature of these relationships. Second, the use of self-reported measures could potentially introduce biases, including social desirability bias and recall bias. Third, this study measures students perceived social support by employing the MSPSS, which solely captures one's general perceptions, disregarding the positive and negative aspects of the social support one perceives. Finally, we use single-item questions with binary options to measure suicidal thoughts among students rather than using a validated instrument. We also did not assess students' suicidal plans or attempts, which limits this study's ability to get a complete picture of the severity of suicidal behaviors among students in a crisis moment. Limitations aside, our findings contribute to a greater understanding of the relationship between perceived social support, adverse mental health outcomes, and suicidal thoughts, thereby offering implications for enhancing the mental well-being of university students.

## Conclusions

Our study revealed a negative association between perceived social support and suicidal thoughts, indicating that higher levels of perceived social support are associated with a reduced likelihood of experiencing suicidal thoughts. Additionally, our findings suggest that perceived stress, distress, anxiety, and depression play a mediating role in the perceived social support and suicidal thoughts relationship. These findings underscore the need for understanding the complex relationship between these constructs to formulate potential intervention points. Since the lower level of perceived social support increases the likelihood of suicidal thoughts by influencing their mental health conditions, suicide prevention efforts at the university level should consider providing social support and psychological help to the students, especially those found to have specific mental disorder symptoms and a feeling of social disconnected- ness. Nevertheless, further investigation based on longitudinal data from diverse contexts is warranted to validate these findings and examine more intricate models that could facilitate the examination of reciprocal effects among all the variables under consideration.

## Supporting information

**S1 Data.**
(SAV)

## Acknowledgments

The authors are grateful to the students who voluntarily participated in the survey. We would like to express our sincere gratitude to the faculty members of various departments at the University of Rajshahi for helping in the data collection process.

## Author Contributions

**Conceptualization:** Sihab Howlader.

**Data curation:** Sihab Howlader.

**Formal analysis:** Sihab Howlader, Sumaiya Abedin.

**Methodology:** Sumaiya Abedin, Md. Mosfequr Rahman.

**Supervision:** Md. Mosfequr Rahman.

**Writing – original draft:** Sumaiya Abedin.

**Writing – review & editing:** Md. Mosfequr Rahman.

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
