## [Decision Letter · Decision Letter 0]

29 Jan 2024

PGPH-D-23-02566

Association of social support with suicidal thoughts among a young adult university student sample in Bangladesh: The mediating role of distress, stress, anxiety, and depression

Dear Dr. Rahman,

Thank you for submitting your manuscript to PLOS Global Public Health. After careful consideration, we feel that it has merit but does not fully meet PLOS Global Public Health’s publication criteria as it currently stands. Therefore, we invite you to submit a revised version of the manuscript that addresses the points raised during the review process.

Kindly address all the issues raised by the reviewersObserving the length of the title of the manuscript, I am suggesting a modification to the original title which in no way would affect the contents of the manuscript.

We look forward to receiving your revised manuscript.

Kind regards,

Nnodimele Onuigbo Atulomah, PhD

Academic Editor

Journal Requirements:

2. Please provide separate figure files in .tif or .eps format only and remove any figures embedded in your manuscript file. Please also ensure all files are under our size limit of 10MB.

Additional Editor Comments (if provided):

The two independent reviewers appointed to evaluate this manuscript both agreed that the study represents valuable contribution to the field and made recommendations for revisions. However I think that the title is too long and should be modified to "Social-support, distress, stress, anxiety, and depression as predictors of suicidal thoughts among selected university students in Bangladesh". Kindly attend to the recommendations made by the reviewers.

Reviewers' comments:

Reviewer's Responses to Questions

**Comments to the Author**

1. Does this manuscript meet PLOS Global Public Health’s publication criteria? Is the manuscript technically sound, and do the data support the conclusions? The manuscript must describe methodologically and ethically rigorous research with conclusions that are appropriately drawn based on the data presented.

Reviewer #1: Yes

Reviewer #2: Yes

2. Has the statistical analysis been performed appropriately and rigorously?

Reviewer #1: Yes

Reviewer #2: Yes

3. Have the authors made all data underlying the findings in their manuscript fully available (please refer to the Data Availability Statement at the start of the manuscript PDF file)?

Reviewer #1: Yes

Reviewer #2: Yes

4. Is the manuscript presented in an intelligible fashion and written in standard English?

Reviewer #1: Yes

Reviewer #2: Yes

5. Review Comments to the Author

Reviewer #1: The authors have scientifically examined the association of perceived social support with suicidal thoughts among a young adult sample of university students, estimated the degree to which perceived stress, distress, anxiety, and depression may explain the association, and concluded that perceived social support reduces the likelihood of suicidal thoughts among university students and that mental disorders have a partial mediating effect on the association.

The authors were able to determine not only the main association (social support predicting suicidal thoughts) but also the contribution of each potential mediator(stress, distress, anxiety, and depression), both individually and collectively, to suicidal thoughts. This is highly commendable.

The study is timely and globally relevant; when published, it will provide evidence to institutions of higher learning, families, clinicians, and health regulatory agencies to adopt strategies that will intentionally provide social support to students with adverse mental health outcomes and reduce incidences of suicidal thoughts, suicidal attempts, and suicide. The recommendations will also enhance the mental well-being of university students if implemented.

The authors are to address the following:

• There was no indication of the type of survey instrument used(online or paper survey instruments, the authors only reported that 642 randomly selected students were interviewed after giving their consent, but did not indicate how the interview was undertaken.

• The sample size was increased to 642 to strengthen the survey. Noting that the type of survey instrument was not indicated, it was not clear to the reviewer if the 642 participants recruited all gave their responses, and that the response rate was 100%. It is necessary to clarify this because the same number of selected students (642) were included in the analysis.

• While the discussion section is expected to analyze the findings of the study and relate them with those of previously published similar studies, authors should avoid repeating results in the discussion section.

Reviewer #2: The manuscript examined the association of perceived social support with suicidal thoughts among a young adult sample of university students and estimates the degree to which perceived stress, distress, anxiety, and depression may explain the association using survey method. The manuscript is generally well written and clearly presented.

Other general and specific comments are provided below:

1. Originality of value: The manuscript presents an original study which was designed and conducted among young adult university students in Bangladesh. The paper addressed a relevant public health issue of concern, namely suicidal thought and with thought-provoking results.

2. Suitability and soundness of technique: The techniques used for the study are adequate and robust. It utilized a cross-sectional research design and questionnaire.

3. Clarity of Presentation: The manuscript was exceptionally clear in narrative presentation. It is easy to follow and complied with the journal specifications.

4. Areas requiring correction. While the paper is generally well written, some corrections are required, which have been highlighted for the authors' attention below

Recommendation: The paper should be accepted with minor correction

Abstract

The authors should include the information on the instrument, validation, and sampling technique in the abstract.

Methods:

The authors should describe in detail and clearly how they selected the participants from the faculties and the number of participants per departments

6. PLOS authors have the option to publish the peer review history of their article (what does this mean?). If published, this will include your full peer review and any attached files.

**Do you want your identity to be public for this peer review?** For information about this choice, including consent withdrawal, please see our Privacy Policy.

Reviewer #1: **Yes: **Ukamaka Gladys Okafor, PhD

Reviewer #2: **Yes: **Titilayo Olaoye

---

## [Editor Report · Decision Letter 1]

18 Mar 2024

Social support, distress, stress, anxiety, and depression as predictors of suicidal thoughts among selected university students in Bangladesh

PGPH-D-23-02566R1

Dear Dr. Rahman,

We are pleased to inform you that your manuscript 'Social support, distress, stress, anxiety, and depression as predictors of suicidal thoughts among selected university students in Bangladesh' has been provisionally accepted for publication in PLOS Global Public Health.

Best regards,

Nnodimele Onuigbo Atulomah, PhD

Academic Editor

Congratulations. Having followed through with all the recommendations of the reviewers for manuscript revision, I wish to state that this manuscript has fully met the expectations of the Academic Editor and recommend the manuscript for publication.